# Towards Continual Domain Adaption of Vision-Language Models

## Abstract

Large-scale vision-language models have achieved remarkable performance on various downstream tasks. Nevertheless, how to efficiently adapt vision-language models to new data distributions without re-training, *i.e.*, domain incremental learning (DIL) of vision-language models, is still under-explored. Existing DIL methods for single modality are either not applicable to multi-modal settings or need exemplar buffers to store previous samples to avoid catastrophic forgetting, which is not memory-efficient. To address these limitations, we propose an exemplar-free paradigm to improve DIL of vision-language models based on prompt-tuning. We theoretically analyze and decompose the problem into two optimization objectives. Guided by the theoretical insights, we propose a novel framework named **M**ultimodal **C**ontinual **D**omain **A**daptation (MCDA), which incorporates two strategies: Multimodal Domain Alignment (MDA) and Maximum Softmax Gating (MSG). MDA enhances cross-domain performance by aligning visual and language representation spaces, while MSG improves the accuracy of domain identification by gating through Softmax probability. Extensivev experimental results demonstrate that our method outperforms current state-of-the-art approaches.

## 1 Introduction

Large vision-language models have achieved remarkable performance on various downstream tasks (Zhang et al., 2022; Adel et al., 2019; Agarwal et al., 2022; Arani et al., 2021). After large-scale pretraining, these powerful models can make zero-shot predictions without requiring any task-specific training examples. This advantage of large vision-language models is promising for the development of more general-purpose models without further tuning. However, the zero-shot performance on certain tasks suffers from a lack of sufficient relevant image-text pairs in the pre-training corpus. For example, images containing domain shift (Aygün et al., 2022; Bhat et al., 2023; Cao et al., 2021) from one domain (*e.g.*sketch) to another (*e.g.*cartoon) with certain textural descriptions are difficult to collect, even though it's crucial for models' open-domain learning adaptability.

Unfortunately, expanding the knowledge of the vision-language model through re-training from scratch would incur prohibitively high computational costs. One effective way to alleviate the issue is to continually fine-tune or prompt-tune the vision-language model on various domains of data, which is known as domain incremental learning (DIL) (Ardywibowo et al., 2022; Benzing, 2022; Boschini et al., 2022b;a;c). In the DIL setting, the set of classes remains constant, but the domains (data distributions) involved commonly vary a lot in sequence while domain indices are not provided at inference time. The continually learned model can handle any image-text input and can be further used for downstream tasks. Nevertheless, domain incremental learning remains challenging for vision-language models. We find that when learning is performed sequentially on multiple domains, pre-trained vision-language models tend to forget most of the knowledge related to previously learned tasks (a task refers to learning on one domain). The phenomenon is commonly known as catastrophic forgetting.

Despite considerable efforts to apply DIL in single-modality settings, these methods either do not apply to multi-modal settings or require exemplar buffers to store previous samples to prevent catastrophic forgetting, which is not memory-efficient. Different from traditional methods that modify all or a subset of the network parameters or store examples in a buffer, a new paradigm arises for

continual learning by optimizing a limited number of learnable prompts. As a pioneer working under such a paradigm, S-prompt (Wang et al., 2022a) treats the learning of the prompts independently, which leads to the best performance per domain. It replaces the use of expensive buffers by optimizing per-domain prompts. During training time, they calculate centroids for each domain by applying $k$-means on the training image features, which are generated with the fixed pre-trained transformer without any prompts. During inference, KNN is used to identify the nearest centroid to the test image and then add the associated domain prompt to the image tokens for classification. Despite the empirical performance gains observed by S-prompting, there is still a gap between empirical success and theoretical analysis.

In this paper, we theoretically analyze the paradigm of S-prompting and argue that the design of its component is sub-optimal. The theoretical analysis also shows that its performance is largely limited by wrong-domain prediction and out-of-domain prediction. Motivated by the limitation, we propose a novel framework called **M**ultimodal **C**ontinual **D**omain **A**daptation (MCDA) with two strategies: Multimodal Domain Alignment (MDA) and Maximum Softmax Gating (MSG) to tackle the issue. MSG transforms the problem of domain selection into out-of-domain detection. MDA alleviates the problem of catastrophic forgetting of the vision-language model by forcing the alignment matrix to be similar to that of the previous domain.

Extensive experimental results shows that our approach outperforms state-of-the-art methods. Specifically, MCDA outperforms existing advanced methods of L2P and S-liPrompts with the highest average accuracy score of 89.17% and the lowest average forgetting score of -0.17%. This suggests that MCDA is more adept at learning new information without significantly forgetting previously learned knowledge.

In summary, our contributions are as follows:

- **From a theoretical perspective**, we analyze the boundedness of a continual learning process of vision-language models, and a clear theorem is presented in Theorem 1.

- **From a framework perspective**, we propose the MCDA as a novel framework using both vision and language adaptation for enhancing continual learning of vision-language models.

- **From an experimental perspective**, our proposed MCDA archives new state-of-the-art performances on the CDDB-Hard, DomainNet and CORe50 datasets.

## 2 RELATED WORK

**Continual learning** refers to learning scenarios that require models to adapt to a sequence of tasks with varying data distributions. One of the major challenges of continual learning is known as catastrophic forgetting, where models tend to forget most of the knowledge they previously learned after adapting to new data. To tackle catastrophic forgetting, numerous methods have been proposed. Regularization-based methods (Kirkpatrick et al., 2017; Zenke et al., 2017; Aljundi et al., 2018; Chaudhry et al., 2018; Zenke et al., 2017) alleviate catastrophic forgetting by adding explicit regularization terms to balance the old and new tasks. Replay-based methods(Vitter, 1985; Chaudhry et al., 2019; Riemer et al., 2018; Borsos et al., 2020; Caccia et al., 2020) try to approximate and recover previous data distributions. Optimization-based methods(Lopez-Paz & Ranzato, 2017; Zeng et al., 2019; Guo et al., 2022; Kong et al., 2022; Liu & Liu, 2021) focus on designing specific optimization procedures and programs. Architecture-based methods(Xue et al., 2022; Serra et al., 2018; Golkar et al., 2019; Jung et al., 2020; Gurbuz & Dovrolis, 2022) focus on constructing task-specific parameters.

**Domain-incremental learning** is one of the most commonly seen scenarios of continual learning. In the setting of DIL, each task (domain) has the same data label space but different distributions. Task (domain) identities are not available at inference time. DIL is involved in many real-world problems such as autonomous driving, where the vehicle meets varying weather conditions in the wild (Mirza et al., 2022).

**Prompt tuning** methods are different from traditional methods that modify all or a subset of the network parameters or store examples in a buffer, L2P (Wang et al., 2022d) begins a new paradigm for continual learning by optimizing a limited number of learnable prompts. After that, several work

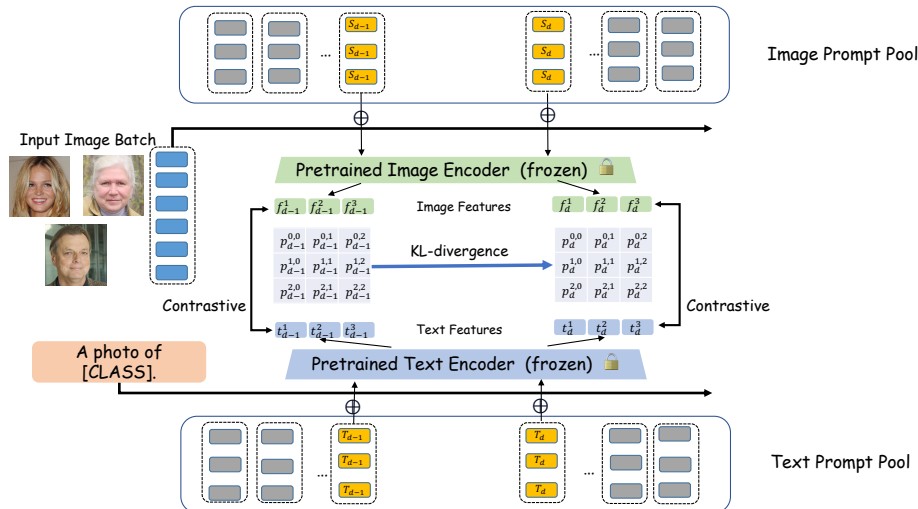

Figure 1: Training Pipeline of MCDA. During the training stage, Multimodal Domain Alignment (MDA) is used to align the image and text representation space between two domains.

(Lester et al., 2021; Li & Liang, 2021; Zhou et al., 2021; 2022; Bahng et al., 2022; Wang et al., 2022c; Douillard et al., 2022a) follow the paradigm and achieve great success.

**Continual learning for vision–Language models** is still under-explored. In (Srinivasan et al., 2022), the focus is on robust fine-tuning of VL. In (Wang et al., 2022b), the change between image and text representation space during the pretraining stage of VL is explored. Recently, ZSCL (Zheng et al., 2023) is proposed to solve the problem of zero-shot degradation during the fine-tuning process of VL. However, little research has been done to explore the potential of prompt-tuning to solve DIL. S-Prompts (Wang et al., 2022a) is the pioneering work to apply prompt-tuning to DIL. It trains different prompts for each domain and dynamically selects the appropriate set during testing using a fixed key/value dictionary. Recently, Hide-Prompt (Wang et al., 2023) decomposes the problem of continual learning into several parts that can be optimized, and derives theoretical analysis of performance. Despite its great success, it focuses on class incremental learning scenarios and could not be directly applied to DIL.

## 3 METHOD

### 3.1 PROBLEM FORMULATION

Denote as $\mathcal{S} = \{\mathcal{D}_s\}_{s=1}^{N}$ as the sequence of datasets presented to the model in our incremental learning scenario. Denote each dataset as $\mathcal{D}_s = \{\mathbf{x}_i^s, \mathbf{y}_i^s\}_{i=1}^{|\mathcal{D}_s|}$, where $\mathbf{x}_i$ represents an image, and $\mathbf{y}_i \in \{0,1\}^K$ is its corresponding one-hot label for $K$ target classes. By convention, in the setting of DIL, we are only allowed to one domain $\mathcal{D}_s$ at a time. Each time a new domain $\mathcal{D}_s$ arrives, the goal of DIL is to improve the model's performance on $\mathcal{D}_s$ and alleviate the catastrophic forgetting for past domains $\mathcal{D}_{s-1}, \mathcal{D}_{s-2}, \ldots, \mathcal{D}_1$.

### 3.2 THEORETICAL ANALYSIS

For domain-incremental learning (DIL), let $\mathcal{X}_t = \bigcup_j \mathcal{X}_{t,j}$ and $\mathcal{Y}_t = \{\mathcal{Y}_{t,j}\}$, where $j \in \{1, \ldots, |\mathcal{Y}_t|\}$ denotes the $j$-th class in task $t$. Now assume we have a ground event denoted as $\mathcal{D} = \{\mathcal{D}_1, \ldots, \mathcal{D}_t\}$ and a pre-trained model $f_\theta$. For any sample $\boldsymbol{x} \in \bigcup_{k=1}^t \mathcal{X}_k$, a general goal of the DIL problem is to learn $P(\boldsymbol{x} \in \mathcal{X}_{*,j} \mid \mathcal{D}, \theta)$, where $\mathcal{X}_{*,j}$ represents the $j$-th class domain in any task. Of note, $\mathcal{Y}_t = \mathcal{Y}_{t'}$, $\forall t \neq t'$ for DIL.

Denote domain identification (DI), within-domain prediction (WDP) and out-of-domain-prediction (ODP) as $P(\boldsymbol{x} \in \mathcal{X}_i \mid \mathcal{D}, \theta)$, $P(\boldsymbol{x} \in \mathcal{X}_{i,j} \mid \boldsymbol{x} \in \mathcal{X}_i, \mathcal{D}, \theta)$ and $P(\boldsymbol{x} \in \mathcal{X}_{k,j} \mid \boldsymbol{x} \in \mathcal{X}_k, k \neq i, \mathcal{D}, \theta)$

respectively. Based on Bayes' theorem, we have

$$P\left(\boldsymbol{x} \in \mathcal{X}_{*,j} \mid \mathcal{D}, \theta\right) = P\left(\boldsymbol{x} \in \mathcal{X}_{i,j} \mid \boldsymbol{x} \in \mathcal{X}_i, \mathcal{D}, \theta\right) P\left(\boldsymbol{x} \in \mathcal{X}_i \mid \mathcal{D}, \theta\right)$$
$$+ \sum_{k \neq i} P\left(\boldsymbol{x} \in \mathcal{X}_{k,j} \mid \boldsymbol{x} \in \mathcal{X}_k, \mathcal{D}, \theta\right) P\left(\boldsymbol{x} \in \mathcal{X}_k \mid \mathcal{D}, \theta\right), \quad (1)$$

where $\{*, j\}$ represents the $j$-th class in each domain. Thus, the problem of DIL can be written as

$$P\left(\boldsymbol{x} \in \mathcal{X}_{*,j} \mid \mathcal{D}, \theta\right) = P_{DI} \cdot P_{WDP} + (1 - P_{DI}) \cdot P_{ODP}. \quad (2)$$

From the formula above, it is clear to see that DIL is constrained by within-domain-prediction, domain identification, and out-of-domain prediction. In other words, the performance of DIL could be improved by enhancing these three objectives.

Within-domain-prediction is related to domain generalization of vision-language models, and there have already been many works trying to improve it. One thing to mention is that our decomposition is similar to HiDe-prompt (Wang et al., 2023) which decomposes the problem of class incremental learning into three objectives. However, in this paper, we focus on domain incremental learning and the decomposition of Hide-prompt does not include ODP. Inspired by HiDe-prompt (Wang et al., 2023), we define

$$H_{\text{WDP}}(\boldsymbol{x}) = \mathcal{H}(\mathbf{1}_{\bar{j}}, \{P(\boldsymbol{x} \in \mathcal{X}_{\bar{i},j} | \boldsymbol{x} \in \mathcal{X}_{\bar{i}}, \mathcal{D}, \theta)\}_j), \quad (3)$$
$$H_{\text{DI}}(\boldsymbol{x}) = \mathcal{H}(\mathbf{1}_{\bar{i}}, \{P(\boldsymbol{x} \in \mathcal{X}_i | \mathcal{D}, \theta)\}_i), \quad (4)$$
$$H_{\text{ODP}}(\boldsymbol{x}) = \mathcal{H}(\mathbf{1}_{\bar{c}}, P(\boldsymbol{x} \in \mathcal{X}_{k,j} \mid \boldsymbol{x} \in \mathcal{X}_k, k \neq i, \mathcal{D}, \theta)), \quad (5)$$

where $H_{\text{WDP}}$, $H_{\text{DI}}$, and $H_{\text{ODP}}$ are the cross-entropy values of WDP, DI, and ODP, respectively. The operation $\mathcal{H}(p, q) \triangleq -\mathbb{E}_p[\log q]$ stands for one-hot encoding function. We now present Theorem 1.

**Theorem 1** *If $\mathbb{E}_{\boldsymbol{x}}[H_{\text{WDP}}(\boldsymbol{x})] \leq \epsilon$, $\mathbb{E}_{\boldsymbol{x}}[H_{\text{DI}}(\boldsymbol{x})] \leq \gamma$, and $\mathbb{E}_{\boldsymbol{x}}[H_{\text{ODP}}(\boldsymbol{x})] \leq \eta$, we have the loss error $\mathcal{L} \in [0, \delta + \epsilon + \log(1 + e^{\epsilon-\delta}(e^{\gamma} - 1))]$*

A detailed proof of the theorem is in Appendix. Theorem 1 shows that optimizing WDP, DI, and ODP can help improve the performance of DIL. In this paper, we focus on improving the last two objectives, namely out-of-domain prediction (ODP) and domain identification, and propose two strategies: Multimodal Domain Alignment (MDA) and Maximum Softmax Gating (MSG). The overall training pipeline is shown in Figure 1.

### 3.3 MULTIMODAL DOMAIN ALIGNMENT

At training time, an independent set of prompts is trained for each domain and is frozen when training on subsequent domain data. At inference time, the sample needs to identify which domain it comes and select the corresponding set of prompts for that domain. If the sample successfully identify the domain it belongs to, then forgetting will not happen. However, if the sample fail to identify the correct domain, then it will select a wrong set of prompts. Therefore, the sample will be tested on a model trained on domain data that are different from its original domain. In other words, if the domain identification process goes wrong, then the model will be tested on out-of-distribution (OOD) data. Intuitively, the performance is expected to drop when a model is tested with OOD data. To validate the statement, we use models prompt-tuned with different domain sources to test on data from different domains and the results are in Figure 2 (a).

From the figure, we can see that no matter which source domain the model is trained on, it will suffer performance drop when tested on OOD data. We hypothesis that the performance drop is caused by the misalignment between the text representation space and image representation space. Formally, define $f(x)$ as the image feature generated by the image encoder $f$, and $\{g(t_j)\}_j^C$ is a set of weight vectors produced by the text encoder $g(.)$ with each $g(t_j)$ representing the text feature of $j$-class. The prediction probability of vision-language model is computed as

$$p\left(y_j \mid x\right) = \frac{\exp\left(\langle f(x), g(t_j) \rangle\right)}{\sum_{k=1}^{C} \exp\left(\langle f(x), g(t_k) \rangle\right)}, \quad (6)$$

| Testing / Training | GauGAN | BigGAN | Wild | Which FaceReal | SAN |
|---|---|---|---|---|---|
| GauGAN | **97.54** | 91.23 | 73.56 | 82.76 | 56.23 |
| BigGAN | 94.12 | **96.81** | 68.54 | 87.55 | 49.98 |
| Wild | 78.97 | 72.43 | **81.32** | 81.25 | 56.87 |
| Which FaceReal | 83.43 | 88.65 | 75.78 | **94.30** | 62.41 |
| SAN | 61.55 | 68.76 | 57.23 | 70.28 | **83.25** |

| Target / Source | GauGAN | BigGAN | Wild | Which FaceReal | SAN |
|---|---|---|---|---|---|
| GauGAN | **0.81** | 0.76 | 0.52 | 0.34 | 0.14 |
| BigGAN | 0.66 | **0.86** | 0.55 | 0.47 | 0.28 |
| Wild | 0.36 | 0.54 | **0.83** | 0.35 | 0.23 |
| Which FaceReal | 0.39 | 0.62 | 0.52 | **0.72** | 0.31 |
| SAN | 0.24 | 0.21 | 0.26 | 0.38 | **0.75** |

(a)                                               (b)

Figure 2: (a) Performance of models prompt-tuned with different domain sources to test on data from different domains; (b) Average cosine similarity of 100 positive image-text pairs from five different domains

where $\langle .,. \rangle$ is the cosine similarity, and $\langle x, y \rangle = \frac{x \cdot y}{|x||y|}$.

From the formula, we see that the prediction process relies on the mutual interaction between image representation space and text representation space. For models prompt-tuned on different domain sources, since both the image and text are prompt-tuned with separate the sets of prompts, it is possible that their image representation space and text representation space are changed synchronously.

To validate the hypothesis, we select 100 positive image-text pairs from five different domains, feed them into models trained on different domain data and get their corresponding image and text representations. Then we calculate their average cosine similarity and the result is shown in Figure 2 (b). We can see from the figure when using encoders trained on different domain data to generate representations for these image-text pairs, the average cosine similarity indeed drops. Therefore, the performance drop is caused by the misalignment between the text representation space and image representation space.

To tackle the issue, we proposed a strategy called Multimodal Domain Alignment (MDA), which can effectively alleviate the misalignment between image and text representation space during training. Specifically, at training time, suppose that we are training on domain data $D_t$. Given the input image-text batch, we first calculate the contrastive matrix $M_t$ using the current set of domain-specific prompts. Then, we calculate the contrastive matrix $O_t$ generated by the original vision-language model without any prompts. Then we force the alignment between these two contrastive matrices using KL Divergence.

$$L_{KL}^t (M_t, O_t) = -\sum O_t \ln \left( \frac{M_t}{O_t} \right). \tag{7}$$

The overall training loss is:

$$L = L_{contras} + \alpha L_{KL}^t, \tag{8}$$

where $L_{contras}$ is the contrastive loss used in regular CLIP training and $\alpha$ is a constant.

### 3.4 MAXIMUM SOFTMAX GATING

At inference time, the test sample needs to identify which domain it comes from and then select the corresponding set of domain-specific prompts. If the sample fails to identify the correct domains it belongs to, then as discussed above, a performance drop will happen. Therefore, improving the domain identification accuracy is another key factor in improving DIL performance under the proposed paradigm. Previous work, such as S-prompt, takes a naive approach by storing prototype centroids at training time and calculating the similarity of testing sample and prototype centroids to select domain at inference time. We argue that this is sub-optimal and that the problem of domain identification can be transformed into the problem of OOD detection.

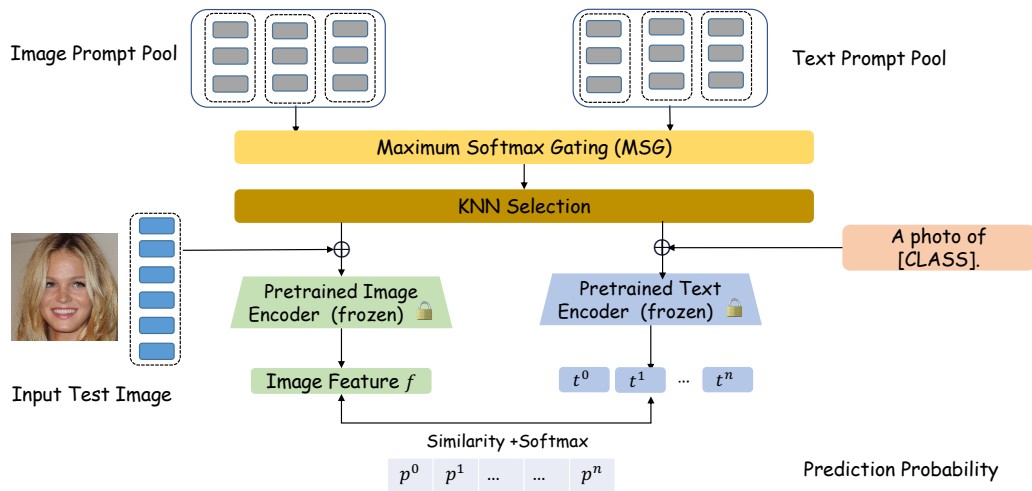

Figure 3: Testing Pipeline of MCDA. During inference stage, Maximum Softmax Gating (MSG) is used to do OOD test of given test image on each domain. Then KNN selection will be performed on domains that survive the OOD tests.

Maximum Softmax Score (MSS) (Ming et al., 2022) is first proposed as a metric for zero-shot OOD detection. We find that MSS is also effective for prompt-tuned vision-language models, thus benefiting the procedure of domain identification. For a given domain with label set $\mathcal{Y}_{\text{in}} = \{y_1, y_2, \ldots, y_K\}$, the text prototypes are set as $\mathcal{T}(t_i), i \in \{1, 2, \ldots, K\}$, where $t_i$ is the text prompt. Then, image and text features are generated by models with a corresponding set of domain-specific prompts. For any test input image $\mathbf{x}'$, we can calculate the label-wise matching score based on the cosine similarity between the image feature $\mathcal{I}(\mathbf{x}')$ and the text vector $\mathcal{T}(t_i)$ : $s_i(\mathbf{x}') = \frac{\mathcal{I}(\mathbf{x}') \cdot \mathcal{T}(t_i)}{\|\mathcal{I}(\mathbf{x}')\| \cdot \|\mathcal{T}(t_i)\|}$. Formally, the Maximum Softmax Score (MSS) is defined as:

$$S_{\text{MSS}}(\mathbf{x}'; \mathcal{Y}_{\text{in}}, \mathcal{T}, \mathcal{I}) = \max_i \frac{e^{s_i(\mathbf{x}')/\tau}}{\sum_{j=1}^{K} e^{s_j(\mathbf{x}')/\tau}}, \tag{9}$$

where $\tau$ is the temperature. For in-domain data, it will be matched to one of the text features with a high score. The OOD detection function can be formulated as:

$$G(\mathbf{x}'; \mathcal{Y}_{\text{in}}, \mathcal{T}, \mathcal{I}) = \begin{cases} 1 & S_{\text{MSS}}(\mathbf{x}'; \mathcal{Y}_{\text{in}}, \mathcal{T}, \mathcal{I}) \geq \lambda \\ 0 & S_{\text{MSS}}(\mathbf{x}'; \mathcal{Y}_{\text{in}}, \mathcal{T}, \mathcal{I}) < \lambda \end{cases}, \tag{10}$$

where by convention 1 represents the positive class (ID) and 0 indicates OOD. $\lambda$ is chosen so that a high fraction of ID data (*e.g.*, 95%) is above the threshold. At inference time, given test data, we conduct an OOD test on all the domains using the OOD detection above. Ideally, only one domain will accept and the other will reject. If that is the case, then the domain that accepts is the one that the test data belongs to. However, we find that in fact, several domains accept, and others reject. Therefore, we propose Maximum Softmax Gating (MSG) to filter out all the domains that fail the OOD test. For domains that survive the OOD test, we will select among them using the same metric as the S-prompt. The overall procedure is shown in Figure 3.

## 4 EXPERIMENT

### 4.1 DATASETS

We perform experiments on three standard DIL benchmark datasets: CDDB (Li et al., 2023), CORe50 (Lomonaco & Maltoni, 2017), and DomainNet (Peng et al., 2019).

**CDDB** is a dataset for continual deepfake detection, which designs easy, long, and hard tracks. Particularly, we choose the most challenging track (*i.e.*, the Hard track) that requests learning on 5 sequential deepfake detection domains, which are GauGAN, BigGAN, WildDeepfake, WhichFaceReal, and SAN respectively.

Table 1: **Results on CDDB-Hard.** Evaluation of existing state-of-the-art DIL methods in the standard DIL setting. The best results are highlighted in **bold**.

| Method | Prompts | Buffer size | AA (↑) | AF (↓) |
|---|---|---|---|---|
| LRCIL (Pellegrini et al., 2020) | ✗ | | 76.39 | -4.39 |
| iCaRL (Marra et al., 2019) | ✗ | *100ex/class* | 79.76 | -8.73 |
| LUCIR (Hou et al., 2019) | ✗ | | 82.53 | -5.34 |
| LRCIL (Pellegrini et al., 2020) | ✗ | | 74.01 | -8.62 |
| iCaRL (Marra et al., 2019) | ✗ | *50ex/class* | 73.98 | -14.50 |
| LUCIR (Hou et al., 2019) | ✗ | | 80.77 | -7.85 |
| DyTox (Douillard et al., 2022b) | ✓ | | 86.21 | -1.55 |
| EWC (Kirkpatrick et al., 2017) | ✗ | | 50.59 | -42.62 |
| LwF (Li & Hoiem, 2017) | ✗ | | 60.94 | -13.53 |
| DyTox (Douillard et al., 2022b) | ✓ | *No buffer* | 51.27 | -45.85 |
| L2P (Wang et al., 2022d) | ✓ | *No buffer* | 61.28 | -9.23 |
| S-liPrompts (Wang et al., 2022a) | ✓ | *No buffer* | 88.65 | -0.69 |
| **MCDA (Ours)** | ✓ | *No buffer* | **89.27** | **-0.07** |

**CORe50** is a widely used dataset for continual object recognition that has 50 categories from 11 distinct domains. The continual learning setting uses 8 domains for incremental training and the rest domains as the test set.

**DomainNet** is a dataset for domain adaptation and domain incremental learning, which has 345 categories and roughly 600,000 images. The images in this dataset are split into 6 domains. The DIL setup on DomainNet is the same as that of S-liprompt. We report the average forward detection accuracy and the average forgetting degree on CDDB-Hard. For CORe50 and DomainNet, we report the average forward classification accuracy.

## 4.2 COMPARISON METHODS

Following S-prompt (Wang et al., 2022a), we benchmark our proposed methods against state-of-the-art DIL methods. These include non-prompting methods: EWC (Kirkpatrick et al., 2017), LwF (Li & Hoiem, 2017), ER (Chaudhry et al., 2019), GDumb (Prabhu et al., 2020), BiC (Wu et al., 2019), DER++ (Buzzega et al., 2020) and Co2L (Cha et al., 2021), prompting-based methods: L2P (Wang et al., 2022d), DyTox (Douillard et al., 2022b) and S-liPrompts (Wang et al., 2022a)and a self-supervised learning method: CaSSLe (Fini et al., 2022).

## 4.3 RESULTS

We evaluate our proposed approach in the standard DIL scenario. Table 1 shows the performance of our method on the challenging CDDB-Hard dataset: Referring to Table 1, we can see that MCDA outperforms all previous state-of-the-art methods. Compared with methods without using prompts, MCDA has clear superiority with an average relative improvement of 17%. Besides, these prompt-free methods normally require exemplar buffers, whereas our method does not. Our method is also superior to prompt-based DyTox by a large margin.

MCDA achieves better performance than DyTox without the use of a buffer, which is more memory-efficient. Compared with recent prompt-based approaches such as L2P and S-liPrompts, our method is still superior. This is largely due to our design of MDA and MSG, which effectively alleviates the forgetting issue of continual domain incremental training of vision-language models and improves the domain selection accuracy.

Table 2 (a) shows the performance of our method on the DomainNet dataset. Compared with the CDDB-hard dataset, DomainNet contains more heterogeneous domains and thus is more challenging to handle domain shift. Due to memory efficiency concerns, we see exemplar-free methods as our real competitors. CaSSLe is an exemplar method that can be used jointly with other self-supervised learning methods and MCDA has nearly 20% performance gain over

Table 3: Results of ablating MDA on CDDB-Hard for exemplar-free deepfake DIL. For a fair comparison, MSG is not applied. $\alpha = 0$: MDA is not used.

| Method | Average Acc (↑) | Forgetting (↑) |
|---|---|---|
| MCDA ($\alpha = 0$) | 88.65 | -0.69 |
| MCDA ($\alpha = 0.1$) | 88.71 | -0.63 |
| MCDA ($\alpha = 0.5$) | **88.95** | **-0.39** |
| MCDA ($\alpha = 1$) | 88.93 | -0.41 |
| MCDA ($\alpha = 5$) | 79.15 | -10.19 |

Table 4: Performance under different domain selection strategies on CDDB-hard. We show that MSG is effective even when it is applied to random domain selection.

| Method | KNN/Random | MSG | AA (↑) |
|---|---|---|---|
| MCDA | KNN | ✗ | 72.43 |
| MCDA | Random | ✗ | 68.57 |
| MCDA | KNN | ✓ | 89.17 |
| MCDA | Random | ✓ | 79.58 |

Table 5: Average domain identification accuracy on CDDB-Hard. We show the domain identification accuracy improvement after applying MSG.

| | GauGAN | BigGAN | WildDeepfake | WhichFaceReal | SAN | Average Acc (↑) |
|---|---|---|---|---|---|---|
| KNN w/o MSG | 0.67 | 0.78 | 0.94 | 0.95 | 0.56 | 0.63 |
| KNN w/ MSG | **0.83** | **0.89** | **0.97** | **0.97** | **0.79** | **0.88** |

CaSSLe. We hypothesize that although CaSSLe achieves memory-efficient domain-incremental learning by using prompts, its performance is restricted by the nature of self-supervised learning.

Compared with L2P, MCDA also achieves a superior performance gain of around 28%. This is because L2P was initially proposed for class-incremental learning. Although the paradigm of L2P can be applied to domain-incremental learning, it cannot handle the domain shift well among different tasks. Contrarily, S-liPrompts, and MCDA both handle the domain shift problem by maintaining a set of separately trained prompts for each domain, which effectively solves the problem of domain shift. MCDA also achieves better performance than S-liPrompts, which is largely due to the design of MDA which regularizes the alignment between visual and language space, and MSG which improves the accuracy of domain selection.

Table 2: **Results on DomainNet.** The results are reported as the Accuracy (Acc) metric, where the best values are highlighted in **bold**.

| Method | Prompt | Buffer size | AA (↑) |
|---|---|---|---|
| DyTox (Douillard et al., 2022b) | ✓ | *50ex/class* | 62.94 |
| LwF 'te'pli2017learning | ✓ | | 49.2 |
| CaSSLe (Fini et al., 2022) w/ SimCLR (Chen et al., 2020) | ✗ | | 44.2 |
| CaSSLe w/ BYOL (Grill et al., 2020) | ✗ | | 49.7 |
| CaSSLe w/ Barlow Twins (Zbontar et al., 2021) | ✗ | *No buffer* | 48.9 |
| CaSSLe w/ SupCon (Khosla et al., 2020) | ✗ | | 50.9 |
| L2P (Wang et al., 2022d) | ✓ | *No buffer* | 40.1 |
| S-liPrompts (Wang et al., 2022a) | ✓ | *No buffer* | 67.7 |
| **MCDA (Ours)** | ✓ | *No buffer* | **70.3** |

Table 7 (b) shows the performance of our method on the CORe50 dataset. Different from the previous two datasets, CORe50 contains unseen tested domains that do not appear in the incremental training stage. This requires the method to be capable of generalizing to unseen domains well. Previous methods, no matter using prompts or buffers or not, fail to take OOD generalization into their design concern. However, MDA in our methods naturally allows for better OOD generalization ability, which is the reason why it outperforms all previous methods on OOD testing domains.

Table 7: **Results on CORe50.** The results are reported as the Accuracy (Acc) metric, where the best values are highlighted in **bold**.

| Method | Prompt | Buffer size | AA (↑) |
|---|---|---|---|
| GDumb (Prabhu et al., 2020) | ✗ | | 74.92 |
| DER++ (Buzzega et al., 2020) | ✗ | *50ex/class* | 79.70 |
| DyTox (Douillard et al., 2022b) | ✓ | | 79.21 |
| L2P (Wang et al., 2022d) | ✓ | | 81.07 |
| EWC (Kirkpatrick et al., 2017) | ✓ | | 74.82 |
| LwF (Li & Hoiem, 2017) | ✓ | | 75.45 |
| L2P (Wang et al., 2022d) | ✓ | *No buffer* | 78.33 |
| S-liPrompts (Wang et al., 2022a) | ✓ | *No buffer* | 89.06 |
| **MCDA (Ours)** | ✓ | *No buffer* | **92.37** |

## 4.4 ABLATION STUDY

**Effect of Multimodal Domain Alignment (MDA).** To validate the effectiveness of our methods, we first conduct an ablation study on the two proposed strategies in MCDA. Table 3 shows the result

Table 6: Average accuracy on OOD domains on CDDB-hard. We show that MCDA enjoys superior domain generalization ability compared with previous methods.

|              | S1    | S2    | S3    | OOD1  | OOD2  | OOD3  | AA    |
|--------------|-------|-------|-------|-------|-------|-------|-------|
| S1 (ours)    | 99.78 | 87.72 | 47.63 | 51.09 | 63.46 | 72.38 | 66.41 |
| S2 (ours)    | 99.92 | 99.21 | 52.31 | 73.56 | 74.38 | 81.50 | 73.26 |
| S3 (ours)    | 99.90 | 98.75 | 82.16 | 73.29 | 66.83 | 78.15 | 75.31 |
| S3 (S-liPrompts) | 97.83 | 73.21 | 69.21 | 62.43 | 61.87 | 71.42 | 71.53 |

of changing the value of hyperparameter $\alpha$ in the training loss on the CDDB-hard dataset. From the results, we can see that when applying MDA and setting $\alpha$ to be in a proper range, our method outperforms the baseline method. When $\alpha$ is set to 0, this means that we do not apply MDA in the training process. $\alpha$ stands for the balance between learning the distribution of the current domain and forcing the model to align the model with a previous domain. From the results, we can see that if $\alpha$ is too large, then performance will drop. We hypothesize that this is because if the weight assigned to the alignment matrix is too large, then it will affect the original adaptation performance.

**Effect of Maximum Softmax Gating (MSG).**  MSG is proposed to improve the domain identification accuracy. Table 5 shows the accuracy of domain selection when applying MSG on the CDDB-hard dataset. From the results, we can see that after applying MSG, the average domain identification accuracy is improved from 63% to 88%. We find that without applying MSG, the model tends to misidentify the test samples between the GauGAN domain and the BigGAN domain. As shown in Table 5, the domain identification accuracy of these two domains both do not reach 80%. However, after applying MSG, the model will first do an OOD test of each domain before selecting it. Therefore, part of the testing samples that would be misclassified will be filtered out during the OOD test. Consequently, the accuracy of domain selection will be improved.

**Exploration of different domain identification strategies.**  MCDA adopts KNN as the domain selection strategy. At the inference stage, the extracted features of the testing image are used to query the domain using the KNN algorithm. Experiments have shown that MSG can help improve the domain selection accuracy when using KNN as the identification strategy. To explore whether MSG is still effective when using other identification strategies, we conduct experiments when domains are randomly selected during inference. Table 4 shows the between using KNN and random selection. We can see that even under the setting of random selection, MSG can still improve the performance.

**Exploration of generalization capability of the model.**  Following Wang et al. (2022a), we apply the trained MCDA prompts on S1-S3 to out-of-domain OOD1-OOD3 which are 3 unseen domains in the CDDB-hard dataset. We choose S1-S3 to be: GauGAN, BigGAN, WildDeepfake; and choose OOD1-OOD3 to be: FaceForensic++, Glow, StarGAN. From the results in Table 6, we can see that our method can achieve good performance on OOD domains, with an average accuracy gain of 4% compared with Sli-Prompts (which achieves the second-best generalization performance among all methods). This indicates that our method is capable of handling cases when there are unseen domains in the testing stage.

## 5 DISCUSSION AND CONCLUSION

In this paper, we theoretically analyze and decompose the problem of DIL into two optimization objectives. Guided by the theoretical insights, we propose two strategies: Multimodal Domain Alignment (MDA) and Maximum Softmax Gating (MSG). MDA improves the model's cross-domain performance by forcing the alignment between visual and language representation spaces. MSG improves the accuracy of domain identification by gating through softmax probability. Experiments show that our method outperforms existing state-of-the-art methods.

The limitation of the proposed MCDA is mainly in two aspects. On the one hand, MSG could still be improved to enhance the domain selection accuracy. On the other hand, our method is restricted for the setting of DIL. Making MCDA adaptable for all CL setting remains to be explored.

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

# Appendix

## A  PROOF OF THEOREM 1

$$H_{DIL}(x) = H\left(y, \{\mathbf{P}\left(x \in \mathbf{X}_{k,j} \mid D\right)\}_{k,j}\right)$$
$$= -\sum_{k,j} y_{k,j} \log \mathbf{P}\left(x \in \mathbf{X}_{k,j} \mid D\right) \tag{11}$$
$$= -\log \mathbf{P}\left(x \in \mathbf{X}_{k_0,j_0} \mid D\right),$$

where $H_{DIL}(x)$ represents the entropy of the distribution over the sets $\mathbf{X}_{k,j}$, given the data $D$ and the probabilities of $x$ being in these sets. The summation simplifies to a single term based on the highest likelihood for a specific set $\mathbf{X}_{k_0,j_0}$.

$$H_{WDP}(x) = H\left(\tilde{y}, \{\mathbf{P}\left(x \in \mathbf{X}_{k_0,j} \mid x \in \mathbf{X}_{k_0}, D\right)\}_j\right)$$
$$= -\sum_{j} y_{k_0,j} \log \mathbf{P}\left(x \in \mathbf{X}_{k_0,j} \mid x \in \mathbf{X}_{k_0}, D\right) \tag{12}$$
$$= -\log \mathbf{P}\left(x \in \mathbf{X}_{k_0,j_0} \mid x \in \mathbf{X}_{k_0}, D\right),$$

Here, $H_{WDP}(x)$ describes the conditional entropy given that $x$ is in $\mathbf{X}_{k_0}$, evaluating the likelihood across subsets $\mathbf{X}_{k_0,j}$.

$$H_{DI}(x) = H\left(\bar{y}, \{\mathbf{P}\left(x \in \mathbf{X}_{k} \mid D\right)\}_k\right)$$
$$= -\sum_{k} \bar{y}_k \log \mathbf{P}\left(x \in \mathbf{X}_{k} \mid D\right) \tag{13}$$
$$= -\log \mathbf{P}\left(x \in \mathbf{X}_{k_0} \mid D\right),$$

In this case, $H_{DI}(x)$ is the entropy over the sets $\mathbf{X}_k$, which gives the likelihood of $x$ belonging to any set $k$. This again reduces to the most probable set $\mathbf{X}_{k_0}$.

$$H_{ODP}(x) = H\left(\tilde{y}, \{\mathbf{P}\left(x \in \mathbf{X}_{k_i,j} \mid x \in \mathbf{X}_{k_i}, D, k_i \neq k_0\right)\}_j\right)$$
$$= -\sum_{j} y_{k_i,j} \log \mathbf{P}\left(x \in \mathbf{X}_{k_i,j} \mid x \in \mathbf{X}_{k_i}, D\right) \tag{14}$$
$$= -\log \mathbf{P}\left(x \in \mathbf{X}_{k_i,j_0} \mid x \in \mathbf{X}_{k_i}, D\right),$$

This describes the entropy $H_{ODP}(x)$ for the case where $x$ belongs to a different set $k_i$ (with $k_i \neq k_0$), and it evaluates the probability within the subsets $\mathbf{X}_{k_i,j}$.

$$H_{DIL}(x) = H\left(y, \{\mathbf{P}\left(x \in \mathbf{X}_{k,j} \mid D\right)\}_{k,j}\right)$$
$$= -\sum_{k,j} y_{k,j} \log \mathbf{P}\left(x \in \mathbf{X}_{k,j} \mid D\right)$$
$$\leq -\log\left(P_{WDP}P_{DI} + P_{ODP}(1 - P_{DI})\right) \tag{15}$$
$$= -\log\left(e^{-\epsilon}e^{-\delta} + e^{-\gamma}(1 - e^{-\delta})\right)$$
$$= \delta + \epsilon + \log\left(1 + e^{\epsilon-\delta}(e^{\gamma} - 1)\right).$$

Finally, this inequality shows the bound on $H_{DIL}(x)$ by combining the weighted probabilities of $x$ belonging to either $\mathbf{X}_{k_0,j_0}$ (with probability $P_{WDP}P_{DI}$) or $\mathbf{X}_{k_i,j_0}$ (with probability $P_{ODP}(1-P_{DI})$). The resulting expression involves the exponents $\epsilon$, $\delta$, and $\gamma$, providing a closed-form solution.

## B  IMPLEMENTATION DETAILS

We implement the proposed MCDA framework in PyTorch with NVIDIA RTX 4090 GPU. The image encoder is implemented with the architecture of ViT-B/16 and the text encoder is the same as the text encoder in CLIP. The embedding dimension is 768 for both image and text encoder. We adopt SGD optimizer with a momentum of 0.9, an initial learning rate of 0.1, a batch size of 128, and a cosine scheduler. The learning epoches for CDDB dataset is 50, and for DomainNet and CORe5 are 10.

## C  ABLATION ON MORE DATASETS

### C.1  DOMAINNET

Table 8 presents the results of ablating the Multimodal Domain Alignment (MDA) strategy on the DomainNet dataset for exemplar-free deepfake domain incremental learning (DIL). To ensure a fair comparison, the Maximum Softmax Gating (MSG) strategy was not applied in these experiments. The table shows the average accuracy and forgetting metric under different values of the $\alpha$ parameter, which controls the strength of the MDA alignment.

The results reveal that introducing MDA with $\alpha = 0.5$ achieves the best overall performance, with an average accuracy of 69.5% and the lowest forgetting value of -0.39. This indicates that a moderate level of MDA alignment significantly improves cross-domain performance while effectively reducing forgetting. Notably, the performance degrades when $\alpha$ is set to extremes: for $\alpha = 0$, which means no MDA alignment is applied, the average accuracy drops to 67.1%, and the forgetting increases to -0.69. Similarly, setting $\alpha = 5$ leads to a substantial decline in both accuracy (63.9%) and forgetting (-10.19), suggesting that overemphasizing MDA can negatively impact the model's ability to generalize across domains.

In summary, the results demonstrate that MDA is crucial for improving the performance of MCDA in domain incremental learning. The optimal balance is achieved at $\alpha = 0.5$, where the trade-off between accuracy and forgetting is the most favorable.

Table 9 compares the performance of different domain selection strategies on DomainNet, with and without the application of MSG. The two domain selection methods under evaluation are K-NN and random domain selection.

The results show that the KNN domain selection strategy consistently outperforms random domain selection. When no MSG is applied, the KNN-based MCDA achieves an accuracy of 69.8%, while the random domain selection results in a significantly lower accuracy of 65.4%. This suggests that KNN is more effective in identifying domains that can benefit from domain incremental learning.

When MSG is applied, the performance of both strategies improves. Specifically, random domain selection with MSG achieves an accuracy of 66.2%, while KNN with MSG achieves 67.1%. These results indicate that MSG is effective in improving domain selection, even when the selection process is randomized.

Table 8: Results of ablating MDA on DomainNet for exemplar-free deepfake DIL. For a fair comparison, MSG is not applied. $\alpha = 0$: MDA is not used.

| Method | Average Acc (↑) | Forgetting (↑) |
|---|---|---|
| MCDA ($\alpha = 0$) | 67.1 | -0.69 |
| MCDA ($\alpha = 0.1$) | 68.4 | -0.63 |
| MCDA ($\alpha = 0.5$) | **69.5** | **-0.39** |
| MCDA ($\alpha = 1$) | 66.1 | -0.41 |
| MCDA ($\alpha = 5$) | 63.9 | -10.19 |

Table 9: Performance under different domain selection strategies on DomainNet. We show that MSG is effective even when it is applied to random domain selection.

| Method | KNN/Random | MSG | AA (↑) |
|---|---|---|---|
| MCDA | KNN | ✗ | 69.8 |
| MCDA | Random | ✗ | 65.4 |
| MCDA | KNN | ✓ | 67.1 |
| MCDA | Random | ✓ | 66.2 |

## C.2  CORe50

Table 10 shows the results of ablation studies on the CORe50 dataset, focusing on the impact of the Multimodal Domain Alignment (MDA) strategy in exemplar-free deepfake domain incremental learning (DIL). For these experiments, Maximum Softmax Gating (MSG) was not applied to ensure a controlled comparison. The table reports average accuracy and forgetting metrics for different values of the $\alpha$ parameter, which controls the strength of the MDA alignment.

The results indicate that introducing MDA with $\alpha = 0.5$ achieves the best performance, with an average accuracy of 91.33% and the lowest forgetting value of -0.39. This shows that moderate MDA alignment significantly enhances cross-domain performance while reducing forgetting. Interestingly, increasing $\alpha$ to 1 does not lead to further improvement, with a slight drop in average accuracy to 91.32% and forgetting remaining at -0.41. Furthermore, setting $\alpha$ to an extreme value of 5 leads to a drastic performance degradation, with an accuracy of 88.54% and forgetting of -10.19, indicating that excessive MDA alignment hampers the model's ability to generalize across domains. In summary, the ablation study demonstrates that MDA is critical for improving the performance of MCDA in domain incremental learning, with the optimal alignment strength being $\alpha = 0.5$, striking the best balance between accuracy and forgetting.

Table 11 compares the performance of different domain selection strategies on the CORe50 dataset, both with and without the application of MSG. The two domain selection methods evaluated are K-Nearest Neighbors (KNN) and random selection.

When MSG is not applied, the KNN-based domain selection yields the highest accuracy of 91.65%, while random selection results in a lower accuracy of 89.76%. This illustrates the effectiveness of KNN in selecting relevant domains for domain incremental learning. However, when MSG is applied, both strategies see a slight performance decrease, with KNN achieving 89.72% and random selection 88.79%. These results indicate that while MSG is generally beneficial, its interaction with different domain selection methods may vary, and in this case, KNN alone proves more effective than combining it with MSG. The result shows that selecting appropriate domain strategies is crucial, and KNN consistently outperforms random selection. The results also suggest that the additional use of MSG should be considered carefully based on the specific domain selection method being employed.

Table 10: Results of ablating MDA on CORe50 for exemplar-free deepfake DIL. For a fair comparison, MSG is not applied. $\alpha = 0$: MDA is not used.

| Method | Average Acc (↑) | Forgetting (↑) |
|---|---|---|
| MCDA ($\alpha = 0$) | 89.02 | -0.69 |
| MCDA ($\alpha = 0.1$) | 89.97 | -0.63 |
| MCDA ($\alpha = 0.5$) | **91.33** | **-0.39** |
| MCDA ($\alpha = 1$) | 91.32 | -0.41 |
| MCDA ($\alpha = 5$) | 88.54 | -10.19 |

Table 11: Performance under different domain selection strategies on CORe50. We show that MSG is effective even when it is applied to random domain selection.

| Method | KNN/Random | MSG | AA (↑) |
|---|---|---|---|
| MCDA | KNN | ✗ | 91.65 |
| MCDA | Random | ✗ | 89.76 |
| MCDA | KNN | ✓ | 89.72 |
| MCDA | Random | ✓ | 88.79 |

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

# A APPENDIX

You may include other additional sections here.

