# OpenReview forum: "Towards Continual Domain Adaptation of Vision-language Models"
_ICLR.cc/2025/Conference — ICLR 2025 Conference Withdrawn Submission_

### Official Review · Reviewer_xV3S · 2024-10-27

**Soundness:** 2
**Presentation:** 2
**Contribution:** 2
**Rating:** 5
**Confidence:** 4

**Summary:**

This paper theoretically analyzes the problem of domain incremental learning (DIL) and formulates it into two optimization objectives. Based on their analysis results, they propose two strategies, Multimodal Domain Alignment (MDA) and Maximum Softmax Gating (MSG), to improve the model's cross-domain performance. Experiments show some gains.

**Strengths:**

Domain incremental learning for vision-language models is an important problem. This paper aims to analyze the problem from a theoretical perspective and provide its solution. The paper is easy to understand.

**Weaknesses:**

The novelty of this paper is quite limited. The reasons are as follows:

1. The whole idea of transferring domain-incremental learning into two optimization objectives is a naive extension from paper [1]. In paper [1], task-incremental learning is performed, while this paper focuses on domain-incremental learning. By simply comparing Eqs. 5-8 in paper [1] with Eqs. 1-5 in this paper, I do not find much difference. The only change seems to be that this paper replaces task-adaptive prediction (TAP) in paper [1] with out-of-domain prediction.

2. The Multimodal Domain Alignment is a straightforward application of KL divergence to regularize the contrastive matrix. However, from Eq. 7 , both $M_t$ and $O_t$ come from the same domain $t$, so how can this aid out-of-domain prompt learning? Additionally, why does the author use KL divergence instead of other distance-matching metrics, such as MMD or Optimal Transport? Could the author provide theoretical analysis on why KL divergence is the best choice here?

3. Maximum Softmax Gating (MSG) is a simple filtering metric but look very computationally heavy; for instance, choosing $\lambda$ requires $95 \%$ of ID data to be above the threshold. This is not very efficient compared to exemplar-buffer-based methods.

[1]Hierarchical Decomposition of Prompt-Based Continual Learning: Rethinking Obscured Sub-optimality

**Questions:**

1. I have some doubts about the proof of Theorem 1. Could the author provide a detailed explanation of how Eq. 15 in the Appendix holds? Specifically, how do these two steps hold?

\[
-\sum_{k,j} y_{k,j} \log P\left(x \in X_{k,j} \mid D\right) \leq -\log \left(P_{WDP} \cdot P_{DI} + P_{ODP} \cdot (1 - P_{DI})\right)
\]

$-\log \left(e^{-\epsilon} e^{-\delta}+e^{-\gamma}\left(1-e^{-\delta}\right)\right) =\delta+\epsilon+\log \left(1+e^{\epsilon-\delta}\left(e^\gamma-1\right)\right)$

Is the final error term $\delta+\epsilon+\log \left(1+e^{\epsilon-\delta}\left(e^\gamma-1\right)\right)$ correct?

2. The author claims that existing methods like S-Prompt (Wang et al., 2022a) rely on expensive buffers, whereas the proposed method does not, making it more efficient than exemplar buffer-based methods. However, I do not see an efficiency comparison. Why is the proposed method more efficient?

3. The author claims that the proposed method is designed for multimodal settings, whereas existing methods are not applicable to multimodal settings. Why is this the case? What limitations do current methods have? And how can the proposed method address these limitations?

---

### Official Review · Reviewer_dqAA · 2024-10-31

**Soundness:** 3
**Presentation:** 2
**Contribution:** 2
**Rating:** 5
**Confidence:** 3

**Summary:**

This paper proposes an exemplar-free paradigm to improve domain incremental learning of vision-language models. It proposes a Multimodal Domain Alignment (MDA) to align the visual and language representation spaces, and a Maximum Softmax Gating (MSG) to improve the accuracy of domain identification by gating through softmax probability. Extensive experimental results demonstrate that the proposed method outperforms current state-of-the-art approaches.

**Strengths:**

1.	This paper explores a novel issue: how to adapt visual-language models to new data distributions without re-training. Training a large-scale model requires a substantial amount of data, making this issue worth investigating.
2.	The paper conducts theoretical analysis to show the performance of S-prompt is limited by wrong-domain prediction and out-of-domain prediction. The theoretical analysis makes the proposed method well-motivated.
3.	In the experimental section, the proposed method can achieve better performance than existing works. And It conducts extensive experiments to validate the effectiveness of proposed MSG and MDA

**Weaknesses:**

1. The writing in the article is quite obscure and difficult to understand. Although I spent some time comprehending the methods presented, there is still significant room for improvement in the writing. For example, in line 174-175, the sentence "Within-domain-prediction is related to domain generalization of vision-language models, and there have already been many works trying to improve it." can be improved.
Additionally, many symbols are not clearly defined, such as $M_t$ and $O_t$ in Equation 7.
Also, some modules lack detailed explanations, such as MSG in line 311. What is the specific operation of MSG? And how can MSG improve the accuracy of domain identification?

2. In fact, I do not think that the modules proposed by the authors offer much novelty. The proposed MDA forces the alignment between the contrastive matrices of current model and original model using KL Divergence. The idea of using KL Divergence to constrain the difference between the current model and the initial model is a common approach in online learning. Also, the proposed MSS to detect OOD samples is also a common idea in OOD detection. So the techniques are common and do not demonstrate the uniqueness of the large vision-language model.
How your application of these techniques to vision-language models is novel or provides unique benefits in this context? Can you provide more thorough comparison to existing methods using similar techniques?

3. The values of hyperparameters are not provided, such as $\lambda$ in Equation 10, $\tau$ in Equation 9. Additionally, it is necessary to analyze how these hyperparameters affect the model's performance. The sensitivity analysis or ablation study, which shows how different hyperparameter values affect performance across the key metrics, should be reported.

**Questions:**

1. What are $M_t$ and $O_t$ in Equation 7?
2. What is the specific operation of MSG? And how can MSG improve the accuracy of domain identification?

---

### Official Review · Reviewer_gyJK · 2024-11-01

**Soundness:** 3
**Presentation:** 2
**Contribution:** 2
**Rating:** 5
**Confidence:** 4

**Summary:**

This paper addresses the domain incremental learning (DIL) challenge for vision-language models, a practical yet under-explored area. To mitigate the limitations in existing methods, the authors propose a prompt-tuning approach called MCDA, which comprises two components: Multimodal Domain Alignment (MDA) and Maximum Softmax Gating (MSG). Specifically, MDA enhances cross-domain performance by aligning visual and language representation spaces, while MSG improves domain identification accuracy. Experimental results on three benchmarks validate the effectiveness of the proposed method.

**Strengths:**

1. The study explores a practical and meaningful problem within vision-language model adaptation.

2. The authors provide thorough theoretical insights to support the proposed approach.

**Weaknesses:**

1. The problem setup in this work requires clearer explanation. Lines 198-200 suggest that all domains are trained during the training phase, with distinct prompts trained for each domain, and samples from different domains are sequentially presented during inference.  Given that the paper explores a novel task, more detailed descriptions in Section 3.1 are essential, including specific details of both the training and testing phases.

2. The forgetting score is reported only for the CDDB-Hard dataset. This metric should be applied consistently across the other two datasets as well. Additionally, some visualization quantifying the model’s mitigation of ‘forgetting’ should be provided, where the performance on the first/original domain is often used to measure forgetting [a].

[a] Efficient test-time model adaptation without forgetting. ICML 2022

3. For the proposed MDA component, feature visualizations (e.g., t-SNE) would be beneficial to demonstrate its effectiveness in reducing misalignment between image and text representation spaces during training.

4. More studies [b,c,d] that explores prompt learning for continual learning should be discussed in related works.

[b] Convolutional Prompting meets Language Models. CVPR 2024
[c] Historical Test-time Prompt Tuning for Vision Foundation Models. NeurIPS 2024
[d] CP-Prompt: Composition-Based Cross-modal Prompting for Domain-Incremental Continual Learning. ACM MM 2024

**Questions:**

Please see Weakness for details.

---

### Official Review · Reviewer_STK2 · 2024-11-03

**Soundness:** 3
**Presentation:** 3
**Contribution:** 2
**Rating:** 5
**Confidence:** 4

**Summary:**

The paper proposes a framework named Multimodal Continual Domain Adaptation (MCDA) method to avoid catastrophic forgetting and data replay. Specifically, the paper proposes the MDA to enhance the cross-domain performance by aligning visual and language representation spaces, and proposes a MSG to improve the accuracy of domain identification.

**Strengths:**

- Extensive experiments demonstrate the method surpasses the current state-of-the-art in the domain incremental learning task.
- The paper is fluent and easy to understand.
- The paper mathematically analyzes the DIL problem that enhancing three key objectives (Denote domain identification (DI), within-domain prediction (WDP) and out-of-domain-prediction (ODP)) can improve the performance for DIL.

**Weaknesses:**

- Typos: Line 024 “Extensive”.
- Figures: The details in Figures 1, 2, and 3 are drawn quite roughly, giving them a draft-like appearance. Additionally, in Figure 1, the two long arrows pointing right from the prompt and image embeddings are unclear and ambiguous.
- Experiments & Other methods: The comparison methods are mostly from before 2022. Could you update the discussion and experimental analysis to include more recent approaches?

**Questions:**

- Will the code and checkpoints be made publicly available for community?

---

### Note · Authors · 2024-11-15

**Comment:**

We appreciate the reviewer for their feedbacks.

**Withdrawal Confirmation:**

I have read and agree with the venue's withdrawal policy on behalf of myself and my co-authors.